# On the Influence of Cavitation Volume Variations on Propeller Broadband Noise

Leonie S. Föhring [1,2,*], Peter Møller Juhl [2] and Dietrich Wittekind [3]

1   Faculty of Mechanical Engineering, Kiel University of Applied Sciences, Grenzstraße 3, 24149 Kiel, Germany
2   Faculty of Engineering, University of Southern Denmark, DK-5230 Odense, Denmark
3   DW-ShipConsult GmbH, 24223 Schwentinental, Germany
*   Correspondence: leonie.foehring@fh-kiel.de; Tel.: +49-431-210-2593

**Abstract:** Low-frequency broadband shipping noise causes a growing concern for marine fauna together with the demand for noise reduction. Detailed analyses of the cavitation behaviour in the time domain serve as a prerequisite for steps toward quieter propellers. Underwater noise measurement data from the propeller of a full-scale container vessel are used for the analysis of the cavitation volume behaviour. Sequencing methods, polynomial models, and stochastic evaluation methods feeding Monte Carlo simulations of unsteady sheet cavitation are applied to identify the cause of an often observed, yet unexplained similarity in shipping noise spectra. Characteristic features of the volume evolution are identified to enhance the understanding of noise generation by sheet cavitation.

**Keywords:** cavitation noise; broadband propeller noise; sheet cavitation

## 1. Introduction

The underwater noise of worldwide shipping has increased over the last few decades [1,2] and continues to affect marine fauna, namely mammals, by inhibiting their means of communication and navigation [3,4]. Single propellers operating in inhomogeneous wake fields are the most prevalent means of propulsion for ocean-going merchant ships and commonly tend to experience sheet cavitation, the loudest type of cavitation [5]. The reduction of shipping noise is imperative and has caught the attention of authorities, shipping companies, and ship designers. It requires a detailed understanding of the noise-generation mechanisms of propeller cavitations.

Typical noise spectra of the propeller cavitation comprise line spectra representing the blade passing frequency and its harmonics as well as broadband components whose shape forms a characteristic hump around 40–50 Hz which is found in many measurements from different ships [6–8]. The tonal components can clearly be explained by the repetitive nature of cavitation caused by the changing inflow conditions experienced by every propeller blade on passing the so-called 12 o'clock position [9]. They are proof of the quasi-stationary behaviour of sheet cavitation. The existence of broadband components is generally attributed to the varying character of a propeller cavitation from a signal analysis point of view [10,11] and based on experimental observations without provision of the detailed generation mechanism [12]. However, an explanation of its specific shape has not yet been found. The studies conducted to date focus on the analysis of varying pressure signals without taking into account the underlying cavitation volume.

The present study follows the approach to calculate the cavitation volume from a measured sound pressure signal, as first proposed by [7] and recently applied by [13]. This approach is based on the monopole characteristic of sheet cavitation noise as described by [5,14] and confirmed in recent experimental and numerical studies [13,15,16]. Noise measurements (both on-board and experimental) are commonly analysed in the frequency domain, lacking connections to the physical processes, while many studies focusing on

the physics of cavitation are performed without noise measurements. [12] combines both methods in model experiments, but a close view of the cavitation behaviour on the model propellers is lacking.

Analysing the cavitation volume behaviour and its resultant noise in the time domain provides the opportunity to study the physical processes involved in the noise generation as recommended in [17]. The core of the novel strategy was introduced by [13] and continues in the present study; it involves the analysis of the cavitation volume and sound pressure signals of propeller blade passages from a full-scale propeller. The previous work of [13] focussed on the development of methods for signal integration and sequencing based on onboard measurement data from a 3600 TEU container vessel provided by [8]. It culminated in the extraction of representative signals of volume and sound pressure for a propeller blade passage and the identification of two variation causes. The resultant signals are in agreement with the cavitation evolution reported in other studies [15,18,19].

This study takes the analysis of the full-scale measurement data and the derived blade passage signals further with a focus on the generation of broadband spectral components. It identifies characteristic shape parameters of the cavitation volume evolution that contribute to the high-pressure peaks on the one hand and create the above-mentioned broadband hump at low frequencies on the other. These shape parameters are set up in modelling the mean signals from [13] with a fitted high-degree polynomial function, tested systematically for their effect on the resultant noise spectrum, and quantified by evaluating the blade passage sequences from the onboard measurements. The application of their mean and standard deviation values within a Monte Carlo simulation of artificially varied noise produces a spectrum similar to the original measurement. This indicates that the identified parameters are among the decisive influencing variables causing the characteristic noise spectra of propeller cavitations and may provide access to their future noise reduction.

## 2. Materials and Methods

### 2.1. Nomenclature and Software

The following sections deal with different types of pressure and cavitation volume time signals in order to distinguish the indices that are used to address each entity individually. Signals and signal sequences from the original pressure measurement are indexed with $m$, such as a continuous signal $V_m(t)$ or the i$^{th}$ sequence $V_{m,i}(t)$ cut from the original measurement. The volume signal modelled with the polynomial function described in Section 2.3 is denoted with $p$ and with an additional $^*$ for the randomly manipulated sequences, e.g., $V_{p,i}^*$. The control points of the polynomial function as pairs of time and volume $(t_j, V_{c,j})$ are denoted with $c$ and numbered with $j$. The mean volume signal shown in Figure 1 is referred to as $\bar{V}(t)$.

All calculations and analyses are performed with MATLAB® version R2021b.

### 2.2. Data Basis

The analysed sound pressure time signals were recorded during an Atlantic passage of a 3600 TEU container vessel whose particulars are given in Table 1 [8]. The measuring conditions are listed in Table 2 specifying the two operating points used for the present study. The sound pressure was measured by a pressure sensor (PGMCA-200KP, Kyowa Electronic Instruments Co., Novi, MI, USA) mounted in the hull plating at the propeller plane on the vessel's centre line with a sampling frequency of 2560 Hz and a measurement duration of 120 s each.

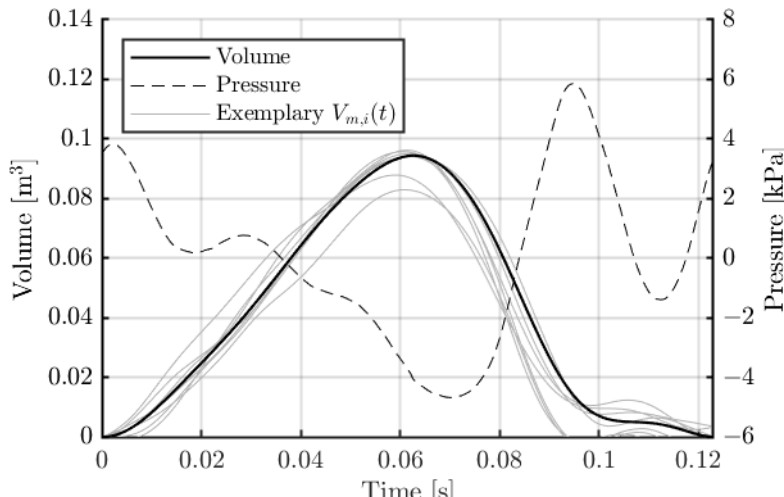

**Figure 1.** Mean signals of cavitation volume and corresponding sound pressure for one blade passage at 96.5 rpm calculated by the *max−L* method as reported in [13] from the total amount of isolated blade passages. Exemplary sequences are included as indicators for the overall variability.

**Table 1.** Vessel characteristics.

| | |
|---|---|
| Length between perpendiculars | 223.57 m |
| Breadth | 32.20 m |
| Maximum draught | 12.50 m |
| Displacement | 63,581 t |
| Engine rating at 104 rpm | 31,710 kW |
| Propeller area ratio | 0.732 |
| Propeller mean pitch | 0.936 |
| Propeller diameter | 7.75 m |
| Number of propeller blades | 5 |
| Hull tip clearance | 2.45 m |

**Table 2.** Measuring conditions.

| | | |
|---|---|---|
| Nominal shaft speed | 96.5 rpm | 87.5 rpm |
| Ship speed (over ground) | 22.2 kn | 20.1 kn |
| Wind force | 2–3 Bft | |
| Sea state (WMO sea state code) | −2 | |
| Water depth | >1000 m | |
| Draught forward / aft | 10.1 m/10.4 m | |

For each of the two operating points, the mean signal of sound pressure and cavitation volume are deduced as described in [13]. Equation (1) relates the volume acceleration to the sound pressure with the seawater density $\rho$ of 1025 kg/m$^3$ and the distance between the sound source and receiver $r$ of 3.8 m. The simplifications of assuming a constant source–receiver distance, neglecting the free water surface, and considering the scattering effect of the hull plating with a solid boundary factor of 1.4, are outlined in [13], and are in agreement with [20]. Integrating the volume acceleration twice, with respect to the time, yields the volume time signal $V_m(t)$.

$$\ddot{V}(t) = \frac{p(t,r) \cdot 4\pi r}{\rho} \tag{1}$$

As proposed in [13], the *max−L* method for the sequencing and aligning of all derived blade passages is applied for the calculation of mean signals for one representative blade passage. The *max−L* method separates the blade passages by identifying the volume maxima which are unambiguously discernible in contrast to the volume minima which

often have two troughs close together. All separated blade passages are stretched to a common signal length and the mean value for each sample point is calculated. The scaling in length helps to average over similar features among all sequences independent from their actual lengths. Thus, the different regions (maxima, collapse, minima) are aligned in order not to distort the mean signal. The resultant volume curve $\bar{V}(t)$ begins at one volume maximum and ends at the next. In order to create the typical volume evolution of a blade passage the mean signal is cut at its lowest point (understood as the junction between adjacent passages). The second part containing the volume increase is set in front of the former part containing the collapse phase to assemble the volume signal as shown in Figure 1. The original pressure signal is treated likewise, resulting in the respective mean signal. Figure 1 also illustrates the strong variability of the individual sequences (light grey).

For the evaluation of individual features of the shape of each volume curve, a different sequencing method *max−cen* is applied, which leaves the length of each passage unchanged. As the duration of the different blade passages is an evident feature of the cavitation evolutions it needs to be preserved. Similar to the previously described method the *max−cen* method separates the volume signal at the maxima. Thus, the collapse process, already identified as significant for the noise generation [16] and located between these maxima, remains unaffected by the separation. Figure 1 shows the highest pressure peaks in the range of the collapsing volume.

### 2.3. Analytical Model for Systematic Manipulation

The evolution process of a cavity growing and collapsing on a propeller blade has both a repetitive and stochastic nature. So, the first objective of this study is to create a parametric model which takes both aspects into account when simulating a continuous volume signal. The repetitive aspect is represented by the use of the mean signal which ensures that all sequences bear the similarities of a common source. The stochastic nature of cavitation behaviour is taken into account by applying the Monte Carlo method in form of randomised variations to the initial model. By creating a continuous volume signal composed of varied blade passages the effect of stochastic variations on the noise spectrum shall be investigated.

Several requirements are placed on a model that represents the mean volume signal shown in Figure 1. As it is primarily a means to investigate the noise generation of cavitation it is essential that the pressure signal can be correctly deduced from it. This requires the second derivative of the function in question to represent the mean pressure signal with sufficient accuracy. Further, the parameters defined in the previous section need to be embedded in a manipulable way to establish their effect on the noise spectrum.

Preliminary analyses [21,22] show that Gauß and Cauchy distribution functions which resemble the overall shape of the volume evolution well-fulfill the first requirement regarding the differentiability, but lack the possibility of manipulating the collapse region sufficiently. Therefore, the presented model is based on a 14-degree polynomial function instead. The high degree is necessary firstly to represent the volume curve in detail and secondly to ensure an accurate fit of the second derivative presenting the pressure signal; 15 control points are spread within the time span of the mean signals as well as outside of it. The former serves to identify significant features, such as the start, maximum, and end of the curve, and flatten the flanks of growth and collapse. The latter resembles the adjacent volume curves with identical shapes to adjust the curvature at the signal ends. Two control points $t_{10}$ and $t_{11}$ are chosen in accordance with the evaluation parameters listed in the next section. Figure 2 shows the original mean signals and the modelled volume curve with its derived pressure signal. The points $t_{10}$ and $t_{11}$ are marked by the black rectangle.

Since the model's objective is to be manipulated and lined up in a new artificial continuous signal, special attention has to be paid to the junction of adjacent sequences in order to avoid discontinuities in the resultant pressure signal. The first measure is the definition of control points before and after the sequence time itself (t < 0 s and t > 0.123 s in

Figure 2). Further smoothing of the junctions is described in Section 2.5 as it is implemented in the setup of the Monte Carlo simulations.

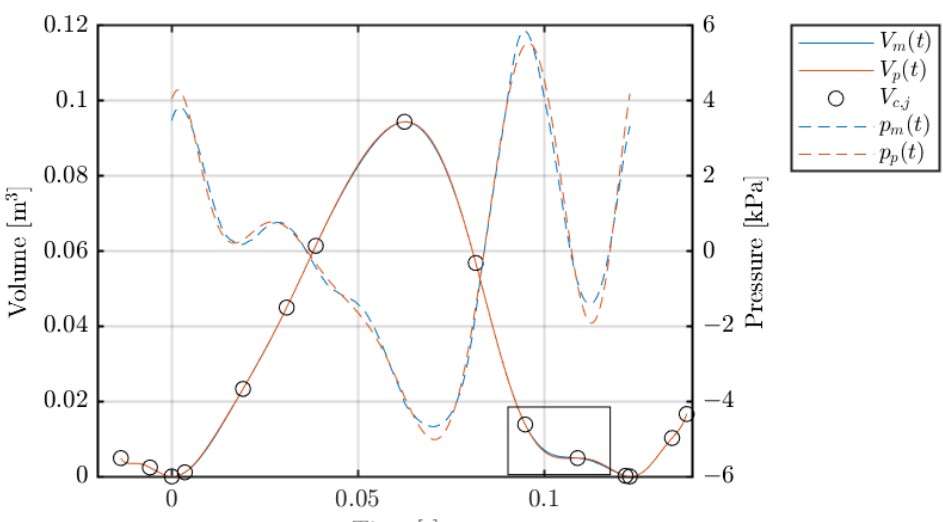

**Figure 2.** Mean signals (blue) as in Figure 1 with the control points (○) for the volume model and the resultant polynomial function with the corresponding sound pressure (orange) for 96.5 rpm.

### 2.4. Evaluation of Shape Parameters

Building on the observation of variations among the large number of individual blade passages [13] the second objective of this study is to identify influencing parameters that determine the sound pressure produced by the growing and collapsing cavity. Figure 1 illustrates the correlation between the curvature (second derivative) of the volume curve and the sound pressure signal defined by Equation (1). Experimental and numerical studies [9] already attributed higher pressure peaks to more rapid collapse events.

In order to grasp this correlation with a geometrically describable entity the position of the pressure maximum $\hat{p}_m$ and the volume minima within each blade passage are analysed. Figure 3 shows their respective position within each sequence. The diagonal line from the origin to the upper right corner marks where pressure maximum and volume minimum occur at the same position within the sequence. Markers below and right from this line represent sequences where the pressure minimum occurs before the respective volume minimum. Each marker represents one blade passage. The passage with a single volume minimum is marked with +. Large amounts of blade passages bear two volume minima close together, as discernible in the exemplary sequences of Section 2.5. In case of these double-volume minima, the positions of both minima are considered (○ and ·), those passages produce two markers accordingly but are still counted only once for the histograms. The proximity of the double minima is visible in the horizontal clustering of the respective markers (○ and ·). Note that the blade passages with only one volume minimum often bear some kind of plateau within the collapse region just before the single volume minimum as shown in the black rectangle of Figure 2. This causes the position of the single minimum to coincide with the position of the second of two minima from other passages. Independent from the number of volume minima the blade passages cluster (vertically in Figure 3) into three main groups depending on the position of the pressure maximum:

(A)  $\hat{p}_m$ at $t > 0.05$ s: pressure peak occurs at the beginning of the subsequent bubble growth
(B)  $\hat{p}_m$ at $0.02$ s $< t < 0.05$ s: pressure peak occurs near the end of the collapse phase
(C)  $\hat{p}_m$ at $t < 0.02$ s: pressure peak occurs near the volume maximum (beginning of sequence)

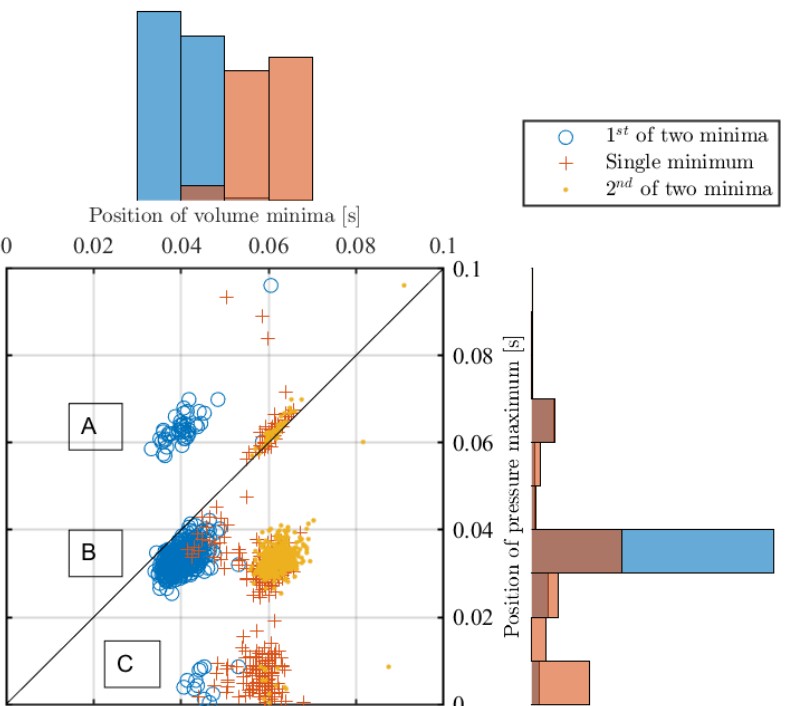

**Figure 3.** Position of the pressure maximum in relation to the position of the volume minima for all passages at 96.5 rpm. The passages are distinguished by the number of volume minima (1 or 2—upper histogram) and by the position of the pressure maximum (groups A, B, and C—right-hand histogram). The colour code of the histograms refers to the marker colour of the centre plot.

For 96.5 rpm, group B is the largest by far, containing most of the individual passages and thus dominating the mean signals from Figure 1 showing the pressure maximum near the end of the large bubble's collapse. For 87.5 rpm, the distribution into the three groups in not as distinct, see Figure 4. Group A contains roughly half of the passages. While the occurrence of the pressure maximum in course of the bubble collapse (group B) agrees with reported observations, a physical explanation for the pressure maximum at the beginning of the bubble growth (group A) is lacking. Therefore and with a focus on the finding for 96.5 rpm, passages from group B are chosen for the stochastic evaluation.

The distinct separation between passages with a single or a double volume minimum in Figures 3 and 4 provides an additional criterion for the identification of shape parameters. The upper histograms show that the two types occur equally often. Figure 5 shows that passages with a double minimum tend to produce higher pressure peaks than those with a single minimum. A similar distribution is found for 87.5 rpm. This emphasises the influence of the collapse region that is outlined in Section 3.1. Additional to the influence of the number of volume minima, it allows also the conclusion that an increasing volume amplitude itself also tends to increase the corresponding pressure peak [13].

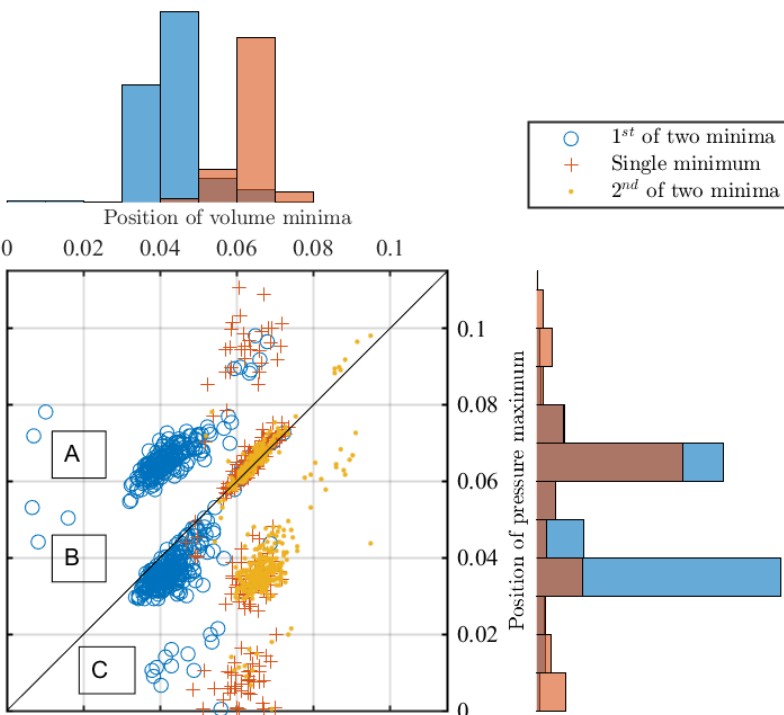

**Figure 4.** Same as Figure 3 for 87.5 rpm.

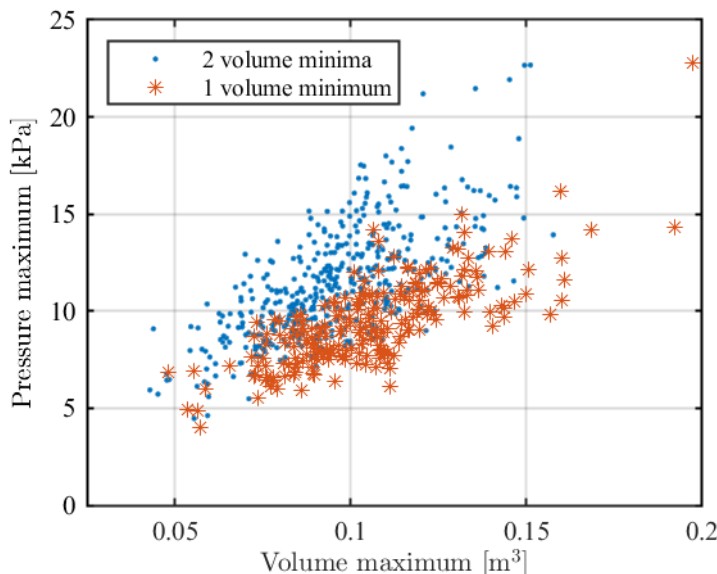

**Figure 5.** Maximum values of volume and pressure within each blade passage of group B at 96.5 rpm distinguished by the number of volume minima.

With regard to the analytic model described in the previous section the following parameters defining the shape of the volume curve are analysed further. The third and fourth parameters are defined to enable the simulation of single or double minima (see the black rectangle in Figure 2).

- Volume amplitude $\hat{V}_{m,i}$, taken at the start of each sequence;
- Length of each sequence $T_{m,i}$, reciprocal to the blade passing frequency;
- Volume at the pressure peak at the collapse region $V_{m,i}(t_{10})$, i.e., the $10^{th}$ control point of the polynomial function;
- Volume in the middle between the pressure peak and the only or second volume minimum $V_{m,i}(t_{11})$, i.e., the $11^{th}$ control point.

These four features are regarded as stochastic random variables

$$X \sim \mathcal{N}(\mu, \sigma^2) \tag{2}$$

where $\mu$ and $\sigma^2$ are calculated from $n$ blade passages by

$$\mu = \frac{1}{n} \sum_{i=1}^{n} x_i \tag{3}$$

$$\sigma^2 = \frac{1}{n-1} \sum_{i=1}^{n} (x_i - \mu)^2 \tag{4}$$

while $x_i$ and $X$ represent the above four parameters for clarity whose calculated values are listed in Table 3. A $\chi^2$–test is performed to ensure the correct assumption of normal distributions. The values of $\mu$ agree with the corresponding features of the mean signal $\bar{V}(t)$, whose polynomial model is used as a basis for the generation of randomly varied sequences (Section 2.5). Only the height of the $11^{th}$ control point is overestimated by the stochastic analysis in comparison to the mean signal. The normalised standard deviation is found to be of equal size in each measuring conditions. The entire process from the raw signal to the manipulable model is displayed schematically in Figure 6.

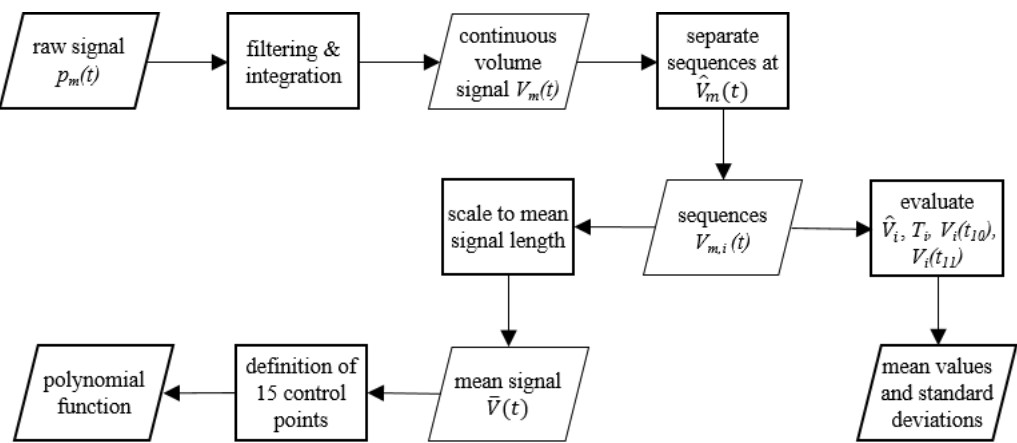

**Figure 6.** Schematic procedure of the data processing from the raw sound pressure signal to the input data of the Monte Carlo simulation. The filtering and integration process is described in detail in [13].

**Table 3.** Stochastic parameters evaluated from blade passage of group B and corresponding features of the respective mean volume signal $\bar{V}(t)$.

| Parameter | | 96.5 rpm | 87.5 rpm | | 96.5 rpm | 87.5 rpm | | 96.5 rpm | 87.5 rpm |
|---|---|---|---|---|---|---|---|---|---|
| Amplitude | $\sigma_a/\mu_a$ | 0.23 | 0.26 | $\mu_a$ | 0.096 | 0.065 | $\hat{V}$ | 0.094 | 0.057 |
| Length | $\sigma_L/\mu_L$ | 0.023 | 0.029 | $\mu_L$ | 0.12 | 0.14 | $\bar{T}$ | 0.12 | 0.14 |
| $10^{th}$ control point | $\sigma_{10}/\mu_{10}$ | 0.55 | 0.43 | $\mu_{10}$ | 0.013 | 0.015 | $\bar{V}(t_{10})$ | 0.0077 | 0.011 |
| $11^{th}$ control point | $\sigma_{11}/\mu_{11}$ | 0.53 | 0.57 | $\mu_{11}$ | 0.0066 | 0.0068 | $\bar{V}(t_{11})$ | 0.0049 | 0.0054 |

### 2.5. Monte Carlo Simulations

In order to understand the mechanisms that lead to the characteristic noise spectrum of propeller cavitations, artificially varied signals are composed by lining up many single sequences created by manipulating the described polynomial function. Each sequence is altered by factors controlling the feature parameters described in Section 2.4. For each parameter, as many random numbers are generated based on a normal distribution described by the standard deviation $\sigma$, such blade passages are to be strung together for the artificial signals. For the purpose of scaling, the modelled signal the stochastic random factors $a$ and $l$ have to be generated around a mean of unity and a standard deviation normalised by the

mean given in Table 3. The process of sequence manipulation and the line-up is shown in Figure 7. The four parameters take effect according to the following equations:

- Volume amplitude–scaling with a constant factor $a_i$

$$V_{p,i}^*(t) = a_i \cdot V_p(t) \tag{5}$$

$$a_i \sim \mathcal{N}\left(1, \frac{\sigma_a}{\mu_a}\right) \tag{6}$$

- Sequence length–scaling and resampling

$$T_{p,i}^* = l_i \cdot T_p \tag{7}$$

$$l_i \sim \mathcal{N}\left(1, \frac{\sigma_L}{\mu_L}\right) \tag{8}$$

- Volume at control point $t_{10}$

$$V_{c,10,i}^*(t_{10}) = a_{10,i} \cdot V_{c,10}(t_{10}) \tag{9}$$

$$a_{10,i} \sim \mathcal{N}\left(1, \frac{\sigma_{10}}{\mu_{10}}\right) \tag{10}$$

- Volume at control point $t_{11}$

$$V_{c,11,i}^*(t_{11}) = a_{11,i} \cdot V_{c,11}(t_{11}) \tag{11}$$

$$a_{11,i} \sim \mathcal{N}\left(1, \frac{\sigma_{11}}{\mu_{11}}\right) \tag{12}$$

The scaling of the sequence length $T_i$ described by Equation (7) is numerically implemented by changing the number of samples contained by the individual sequence. Depending on the factor $l_i$, more or less samples are considered to describe the sequence. Having setup an auxiliary time vector of appropriate length, the original sequence is resampled by interpolation. Since the sampling frequency is assumed to be constant, the reduction of the sample number is effectively the shortening of the sequence and vice versa.

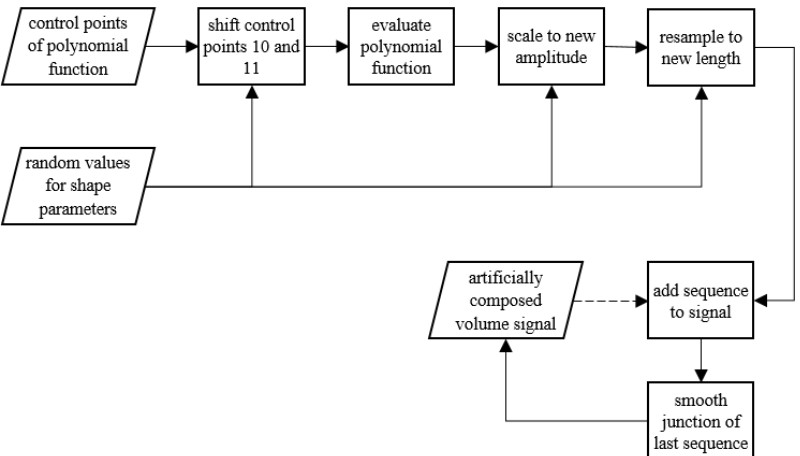

**Figure 7.** Schematic process of the Monte Carlo simulation comprising the individual sequence manipulation and the composition of the artificially varied volume signal.

As mentioned in Section 2.3, the junction between the sequences is a source for discontinuities, especially in the pressure signal which is proportional to the second derivative of the cavitation volume. In order to provide a continuous pressure signal for the spectral analysis, differentiability class $C^2$ is needed for the volume signal. Since the mean signal starts and ends with zero values, which are unchangeable by the scaling according to Equations (5) to (11) only $C^0$ continuity is provided by the lining-up itself. It leads to artificial distortions of the derived pressure signal as visible in the blue line of Figure 8. Ideally, $C^\infty$ would ensure that no additional broadband component is created. However, this would require a significant number of samples of both adjacent sequences to be manipulated to obtain a smooth transition changing the actually desired features of each sequence. Therefore, $C^2$ continuity is chosen as a balance. Adjustments of the curvature (by polynomial control points before and after the actual sequence time span) do not guarantee a smooth transition between the sequences. Therefore, after each amendment of the artificially composed signal by the next manipulated sequence the last four sample points on either side of the junction are deleted and recalculated by spline interpolation for the respective time samples. The spline creates a $C^2$ connection between the volume sequences. The effect of this smoothing is shown in the orange lines in Figure 8.

Despite these efforts, a signal assembled of unvaried sequences still bears remaining junction problems that produce a visible though low broadband component which can be observed in the blue curves shown in all Figures of the next section. By applying a Hanning window [23] to the assembled signal, the effect of a rectangular time window can be reduced to create a spectrum more similar to the ideally clean line spectrum. In the frequency range of interest, the remaining noise floor is approximately 30 dB lower than the broadband noise in the originally measured spectrum (Section 3.2) and is, therefore, assumed not to affect the results presented in the next sections.

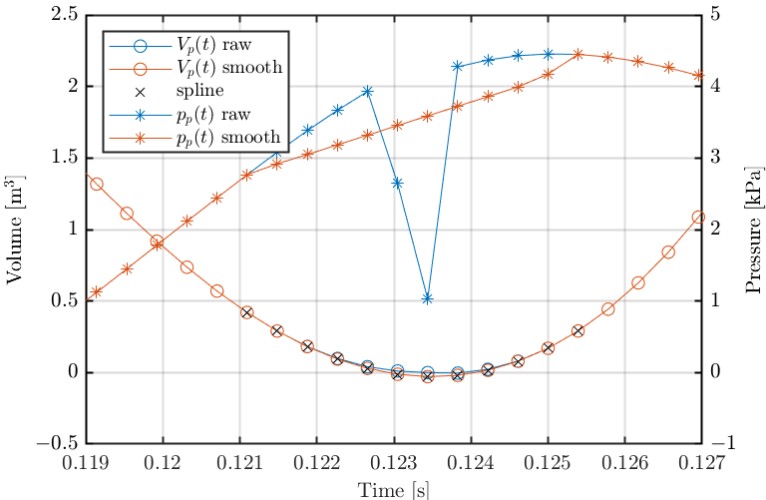

**Figure 8.** Exemplary sequence junction of the volume signal (○) without ('raw') and with smoothing by spline interpolation and the resultant pressure signal (*) showing the involved sampling points (x). No parameter variation is applied to the two sequences.

## 3. Results

### 3.1. Systematically Varied Signal

The generation of artificially varied volume signals described in Section 2.5 is conducted in the first step with only one parameter at a time in order to investigate its individual effect on the noise spectrum (Figures 9–13). The magnitude of the arbitrary values of $\sigma$ is kept comparable to the respective values calculated from the measured sequences. Generally, parameters that change the whole sequence (amplitude and length) tend to have an effect over a wide frequency range, whereas parameters changing only the collapse region of the sequence are observed to have their strongest effect over a limited frequency

range. The results of the systematic parameter variation are outlined on the basis of the 96.5 rpm signal only to avoid redundancy.

Figure 9 shows the resultant spectrum for an isolated variation of the volume amplitude with three different standard deviations $\sigma_a$ controlling the generation of random factors $a_i$. As described in previous studies [11] the amplitude variation generates a broadband component in the spectrum that is roughly constant over the whole frequency range. With $\sigma_a$ increasing the broadband component can be observed to detach itself from the loop-shaped broadband component of the clean signal and follow a line parallel to the harmonics. The peaks of the harmonics of the repetition frequency, i.e., the blade passing frequency remain almost unchanged. Given the significantly lower loops of the clean signal, the effect of the $C^2$ differentiability class of the volume signal at the sequence junctions discussed in Section 2.5 is confirmed to be negligible. The smoothing process by spline interpolation mitigates the patching disturbances sufficiently.

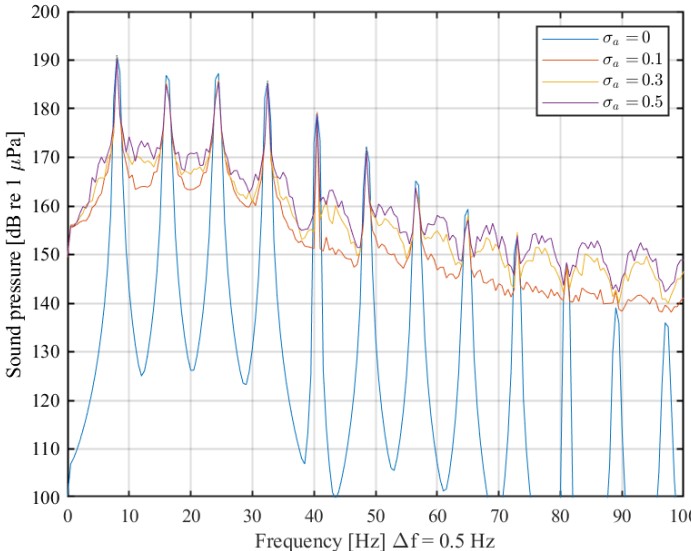

**Figure 9.** Sound pressure spectra of artificial signals generated by variation in the cavitation volume amplitude based on three different standard deviations $\sigma_a$ in comparison to the unvaried signal (blue line) for 96.5 rpm.

The sequences leading to Figure 10 vary only in their length. The continuous signal of the lined-up blade passage corresponds to a varied blade passing frequency, e.g., due to variations of the propeller shaft speed presented in [13]. The initially clean harmonics of the unvaried signal transform into broader peaks with increasing standard deviation. This effect is known as smearing [23]. In particular, the second and higher harmonics resolve increasingly into the rising broadband part of the spectrum, which is also in agreement with the literature. However, the linear increase of the broadband component with the frequency, which is outlined in [11], is difficult to distinguish. The smeared harmonics appear to 'loose' their energy into the surrounding broadband component.

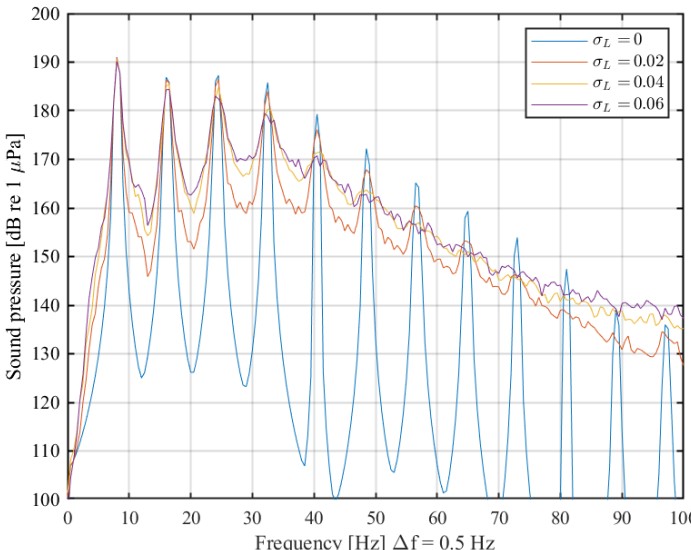

**Figure 10.** Sound pressure spectra of artificial signals generated by variation of the sequence length, i.e., the repetition frequency based on three different standard deviations $\sigma_L$ in comparison to the unvaried signal (blue line) for 96.5 rpm.

Figures 11 and 12 show the individual influence of the two control points defining the collapse region of the volume. Varying only the $10^{th}$ control point of the polynomial function causes the frequency range of approximately 20 to 40 Hz to develop a pronounced broadband hump while leaving the harmonic peaks intact. Only the higher harmonics tend to disappear in the rising broadband component. Varying the $11^{th}$ control point alone causes a similar, though lower hump developing into a slightly lower broadband component in a frequency range above the fifth harmonic.

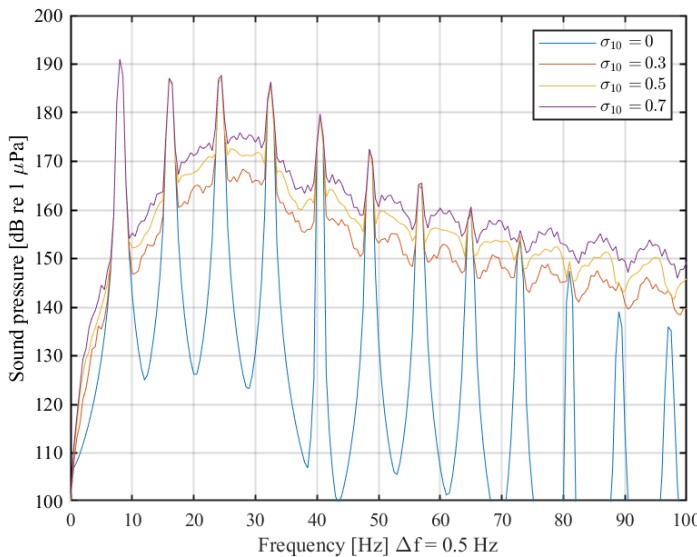

**Figure 11.** Sound pressure spectra of artificial signals generated by variation of the $10^{th}$ control point defining the collapse region based on three different standard deviations $\sigma_{10}$ in comparison to the unvaried signal (blue line) for 96.5 rpm.

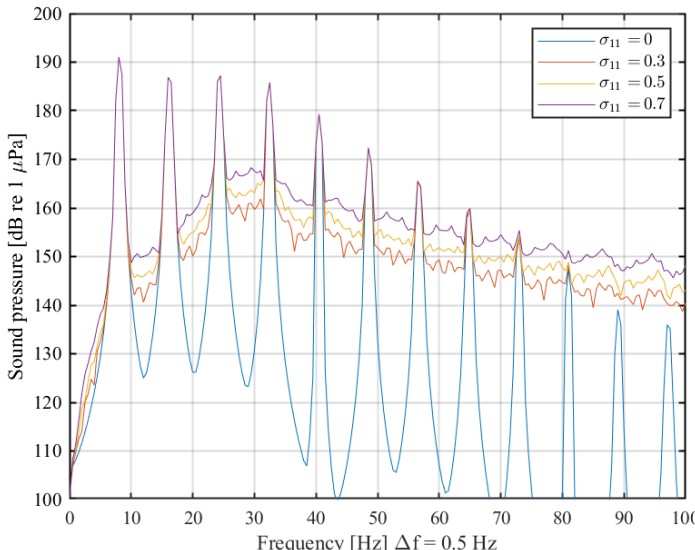

**Figure 12.** Sound pressure spectra of artificial signals generated by variation of the $11^{th}$ control point defining the collapse region based on three different standard deviations $\sigma_{11}$ in comparison to the unvaried signal (blue line) for 96.5 rpm.

The simultaneous variation of both points combines the effects described above. The hump between the second and fourth harmonic becomes more pronounced while harmonics above 70 Hz tend to disappear in the broadband component. The shape of the hump differs clearly from the shape of the broadband component created by the variation of the amplitude or the repetition frequency due to its limitation to the comparatively narrow frequency range of approximately 15 to 40 Hz.

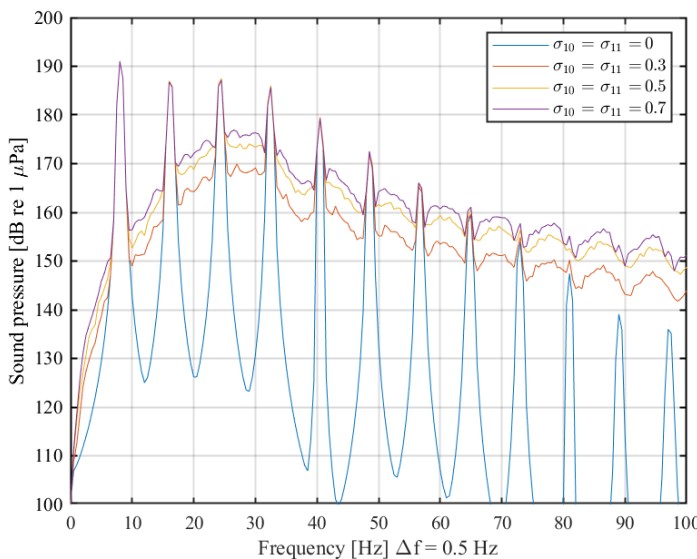

**Figure 13.** Sound pressure spectra of artificial signals generated by simultaneous variation of both control points defining the collapse region based on three different standard deviations $\sigma_{10}$ and $\sigma_{11}$ in comparison to the unvaried signal (blue line) for 96.5 rpm.

### 3.2. Application of Stochastic Parameters from Measured Sequences

The mean values and standard deviations for the four parameters evaluated from the large number of measured blade passages are used for the normal distributions from which the random numbers for the artificially varied signals are generated. The parameters are

taken into account one by one as shown in Figure 14 for 96.5 rpm in order to retrace the effect of each one. The varied parameters are applied to the clean signal in the order listed in Section 2.4. Finally, the original spectrum is presented for comparison.

Figure 14 shows that the two global parameters (amplitude and repetition frequency) which are applied first do not suffice to generate the magnitude of broadband components as present in the original spectrum. Up to the third harmonic of the blade passing frequency a general agreement of the spectra with amplitude and frequency can be observed. On closer examination of the tonal peaks, it can be seen that varying the amplitude alone leaves the high peaks of the clean signal without reduction as stated in the previous section. The additional variation of the repetition frequency provides the necessary reduction by smearing the tonal peaks to reach an agreement with the original spectrum of $p_m$. Above the third harmonic the broadband hump becomes more pronounced and can only be simulated by including the variation of the points defining the collapse region. Between the third and the fourth harmonic, their effect of increasing the broadband component surpasses the original measurement slightly whereas above the fourth harmonic the spectra coincide. Here, the broadband component exceeds the tonal peaks of the clean line spectrum clearly, gaining similarity with the original spectrum again from the variation of the collapse region.

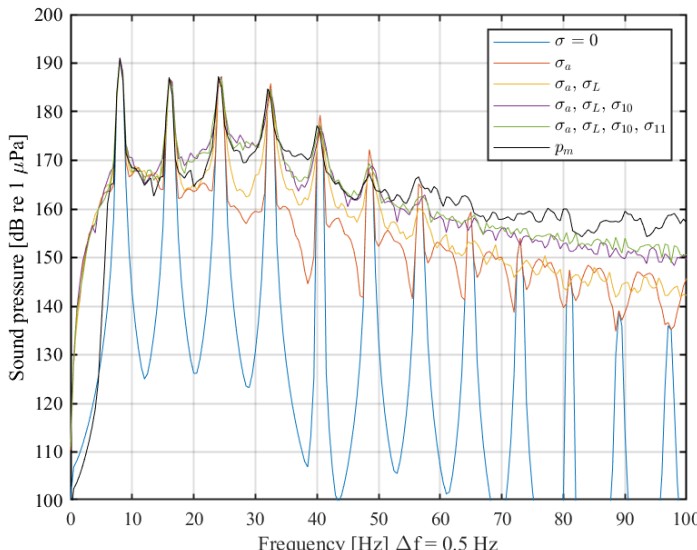

**Figure 14.** Sound pressure spectra of artificial signals generated by variation of all four shape parameters taking effect successively based on standard deviations evaluated from the measured blade passages as stated in Table 3 for 96.5 rpm in comparison to the unvaried signal (blue) and the measured signal $p_m$ (black).

The same procedure is applied to the second measurement (87.5 rpm). Figure 15 shows that the observed effects presented for 96.5 rpm are valid here as well. However, it must be noted that the broadband component generated by the variation of the collapse region exceeds the original spectrum further than for the first operating point. Furthermore, the fifth, sixth, and seventh harmonics are discernible in the original spectrum while they disappear in the broadband component of the artificially varied signal. The reasons for a less close agreement between the artificial and the original spectrum are difficult to establish. Considering the slightly more excessive broadband component between the second and the fourth harmonics, the extent of variations in the collapse region may be overestimated which may correlate with the group B sequences that form only half of the total amount of blade passages (Figure 4, right-hand histogram).

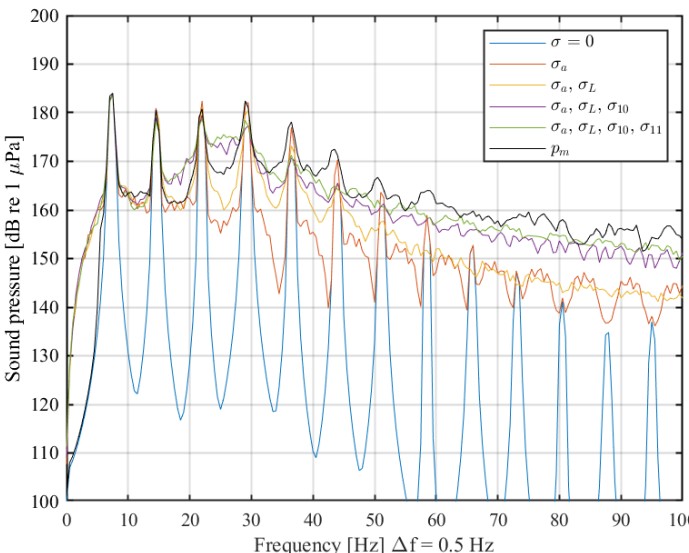

**Figure 15.** Sound pressure spectra of artificial signals generated by variation of all four shape parameters taking effect successively based on standard deviations evaluated from the measured blade passages as stated in Table 3 for 87.5 rpm in comparison to the unvaried signal (blue) and the measured signal $p_m$ (black).

## 4. Discussion

In this study, four shape parameters of the cavitation volume evolution are identified, which can be extracted from a large number of single measured blade passages on the one hand. On the other hand, they are successfully used to manipulate the mean volume signal presented in [13] in course of generating an artificially varied pressure signal. The latter bears a strong resemblance of the spectrum with the originally measured pressure signal and thus confirms that the parameters grasp the variability of the cavitation behaviour well. The mean signal serves as a basis for a parametric analytical model which enables the systematic investigation of the correlation between the cavitation behaviour and the resultant underwater noise.

The variability of the cavitation behaviour becomes apparent in the strong diversity of the isolated blade passages. Even after categorizing them into the groups described in Section 2.4 the considered sequences vary over their whole extent. Therefore, uncertainties remain in grasping the defined shape parameters. The fact that these are still able to simulate spectra similar to the original measurements suggests a significant role for them in the generation of the cavitation noise.

The limitations of this method lie in its experimental simplicity since the cavitation volume evolution that is analysed and manipulated in this study is derived from a measured sound pressure signal and not visually observed. This implies that all measured noise is attributed to the cavitation process by application of the confirmed monopole model of the sound source. Considering the close vicinity of the propeller blade and pressure sensor this assumption appears to be justified given the lack of additional noise sources with equivalent sound levels. However, it cannot be verified by means of independent information on the cavitation volume. Underwater noise measurements onboard full-scale ships require high financial and logistical efforts and are thus rare in literature. Visual observations of full-scale propeller cavitation are even more difficult to conduct and also to analyse quantitatively due to adverse ambient conditions.

Despite these limitations, the present study develops the work on broadband components of cavitation noise [10,11] further by including the physical background to the previous considerations on a varied pressure signal from a propeller cavitation blade. After introducing a relatively simple model of a mean cavitation volume signal, the original

noise spectrum is simulated quite accurately by the stochastic variation of four model parameters. This sheds light on the influence of these features of the volume evolution and indicates that the characteristic but hitherto unexplained hump of the broadband component at approximately 50 Hz can mainly be explained by the strongly fluctuating behaviour of the collapse region. This can be a step toward a better understanding of the relationship between the physical process of propeller cavitations and the measured noise.

In order to substantiate the presented findings, further analyses of measurements onboard other ships with propeller cavitations are needed. Thus, the processes within the cavitation evolution could be studied on a broader basis. The current results may contribute to the overall objective of reducing the cavitation noise by addressing the noise generation mechanisms in more detail. Especially an alleviation of the collapse process may be a step toward noise reduction. The presented method may also be adapted for spectral analysis in combination with numerical simulations of the propeller cavitation.

**Author Contributions:** Conceptualization, L.S.F. and D.W.; methodology, L.S.F.; software, L.S.F.; validation, L.S.F.; formal analysis, L.S.F. and P.M.J.; resources, D.W.; data curation, D.W.; writing—original draft preparation, L.S.F.; writing—review and editing, L.S.F., P.M.J. and D.W.; visualization, L.S.F.; supervision, P.M.J. and D.W. All authors have read and agreed to the published version of the manuscript.

**Funding:** This research received no external funding.

**Data Availability Statement:** The data presented in this study are openly available in Zenodo at DOI 10.5281/zenodo.7404780.

**Acknowledgments:** The analysed raw data of pressure fluctuations were kindly provided by DW-ShipConsult GmbH from its own measurements. We acknowledge the financial support from Land Schleswig-Holstein within the funding programme Open Access Publikationsfonds.

**Conflicts of Interest:** The authors declare no conflict of interest.

## Abbreviations

The following abbreviations are used in this manuscript:

| | |
|---|---|
| Bft | Beaufort scale |
| $p_m$ | measured sound pressure |
| rpm | rotations per minute |
| $T$ | length of one blade passage |
| TEU | twenty foot equivalent unit |
| $\bar{V}$ | mean volume signal |
| $V_c$ | cavitation volume at control point of the polynomial function |
| $V_m$ | cavitation volume calculated from the measured pressure |
| $V_p$ | cavitation volume modelled by polynomial function |
| $V_p^*$ | cavitation volume modelled by manipulated polynomial function |
| WMO | world meteorological organisation |

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
