# Peer review of "On the Influence of Cavitation Volume Variations on Propeller Broadband Noise"

_jmse, doi:10.3390/jmse10121946_

Round 1

Reviewer 1 Report

The paper presents the influence of cavitation volume on propeller noise. Some comments can be considered during the revision as follows:

1. It is not clear the novelty of the work as the authors mentioned several papers that almost have the same concept, in particular, ref 12; please clarify.

2. Does it has a relation with the percentage and distribution of wake field?

3. The introduction needs improvements.

4. The authors can provide a schematic diagram showing the process of computation and data collection to be easily followed and clear to the readers.

5. I think sec 4 can be the conclusion.

6. The authors can add al  the symbol the section of abbreviation

7. The authors can try to compare the results with the empirical formulas used for cavitation from the design point of view, such as Keller and Burrill as well as noise limitations as presented in this paper: https://doi.org/10.3390/jmse10081039

8. Why there is a reduction in Vp(blue line) in fig 7?

9. From your point of view regarding the results, what are the procedures that you suggest to reduce noise and cavitation? You can mention at the end as recommendations

Reviewer 2 Report

Overall, this study has clear research objectives, reasonable research methods, and meaningful research results. However, there are still some issues that need to be raised.

1.The research gap needs to be highlighted

2.The introduction of sensor arrangement is not clear enough.

3.Is there any other interference signal within the signal collected by the pressure sensor?

4.This study focuses on signal analysis, but are the signals themselves representative?

5.It seems more appropriate to submit this paper to a signal processing related journal.

Round 2

Reviewer 1 Report

Accept

Author Response

We appreciate the acceptance of our amendments and thank you for the helpful comments which improved the submitted research report.

Reviewer 2 Report

None

Author Response

(The authors gave the same response as above.)
